# HDL-C Role in Acquired Aortic Valve Stenosis Patients and Its Relationship with Oxidative Stress

**DOI:** 10.3390/medicina55080416

**Published:** 2019-07-29

**Authors:** Juris Hofmanis, Dace Hofmane, Simons Svirskis, Vitolds Mackevics, Peteris Tretjakovs, Aivars Lejnieks, Salvatore Santo Signorelli

**Affiliations:** 1Faculty of Medicine, Department of Internal Diseases, Riga Stradins University and Riga East University Hospital, LV-1007 Riga, Latvia; 2Zemgale Health Center, LV-3001 Jelgava, Latvia; 3Institute of Microbiology and Virology, Riga Stradins University, LV-1007 Riga, Latvia; 4Faculty of Medicine, Department of Human Physiology and Biochemistry, Riga Stradins University, LV-1007 Riga, Latvia; 5Department of Clinical and Experimental Medicine, University of Catania and c/o University Hospital “G.Rodolico”, 95123 Catania, Italy

**Keywords:** aortic valve stenosis, AS, myeloperoxidase, MPO, high-density lipoprotein cholesterol, HDL-C, thioredoxin reductase 1, TrxR1, oxidative stress

## Abstract

*Background and objectives*: Mechanical stress is currently considered as the main factor promoting calcific aortic valve stenosis (AS) onset. It causes endothelial damage and dysfunction. The chronic inflammatory process causes oxidative stress. Oxidative stress-induced high-density lipoprotein cholesterol (HDL-C) dysfunction is an important component of the development of AS. The aim of the study was to evaluate the role of HDL-C in AS patients in three severity grades and in relation to the biomarkers of oxidative stress, thioredoxin reductase 1 (TrxR1) and myeloperoxidase (MPO). *Materials and Methods*: 18 patients with mild, 19 with moderate. and 15 with severe AS were included in the study, and 50 individuals were enrolled in the control group. Stenosis severity was determined by echocardiography. The TrxR1 and MPO were analyzed by ELISA, and HDL-C by commercially available tests. Data were analyzed using GraphPad Prism 8. *Results*: HDL-C in AS patients vs. control substantially decreases and this decline was observed in all three AS severity groups: mild (*p* = 0.018), moderate (*p* = 0.0002), and severe (*p* = 0.004). In both the control and the stenosis group, the HDL-C was higher in women than in men. In comparison to control, the HDL-C level was lower in the AS group, and more pronounced in women (*p* = 0.0001) than in men (*p* = 0.049). A higher TrxR1 level was observed in patients with mild (*p* = 0.0001) and severe AS (*p* = 0.047). However, a clear correlation between TrxR1 and HDL-C was not obtained. Analysis of MPO showed differences in all severity grades vs. control (*p* = 0.024 mild stenosis; *p* = 0.002 moderate stenosis; *p* = 0.0015 severe stenosis). A negative correlation (*p* = 0.047; *rp* = −0.28) was found between MPO and HDL-C, which confirms the adverse effects of MPO resulting in HDL-C dysfunction. *Conclusions*: In this study, we justified HDL-C level association with AS development process. The results unequivocally substantiated the association between HDL-C and AS in all severity grades in women, but only in moderate AS for men, which we explained by the small number of men in the groups. The obtained correlation between the HDL-C and MPO levels, as well as the concurrent decrease in the HDL-C level and increase in the TrxR1 level, indicate in general an HDL-C association with oxidative stress in AS patients.

## 1. Introduction

Calcific stenosis of the aortic valve (AS) manifests with fibro-calcific remodeling (transformation) of the aortic valve (AV), which is a slow process of chronic inflammation and calcification with completely unexplored and ambiguous etiology and pathogenesis [1,2,3,4]. AV sclerosis (thickening of the AV cusps without any flow disturbance) is found echocardiographically (Ehokg) in almost 25% of people after age 65. About 17% of these people develop moderate and severe AS during the next 6–8 years [5]. The prevalence of AS increases with age. Studies on the etiopathogenesis of the disease and the possibility to pharmacologically change the course of the disease are still ongoing. Nevertheless, there is no medical treatment to stop or delay the progression of the disease. The only available treatment is surgical [2]. If surgical treatment is not applied, about 50% of patients die within the next 12–18 months [6,7]. The most frequent complication of surgically untreated and severe aortic stenosis is sudden death [4].

Mechanical stress is currently considered to predominate in AS development. Overall, AS is a chain of complex and highly regulated processes [2]. Mechanical stress causes endothelial damage and dysfunction. Lipid infiltration occurs and inflammatory cytokines are released. Lipids and cytokines further contribute to the endothelial damage, amplifying the inflammatory process. The chronic inflammatory process underlying the progression of AS causes oxidative stress and reduces the capacity of cellular antioxidants. Oxidative stress is an imbalance between reactive oxygen species (ROS) or free radicals and antioxidant protection. Oxidative stress may activate transcription factors, which cause gene expression involved in the inflammatory process. Inflammation caused by oxidative stress promotes many chronic diseases. Here, we see the close relation between both processes: inflammation causes oxidative stress and vice versa. Interaction between ROS, endothelium, and antioxidants is important in the disease development. Superoxide is one of the most active ROS molecules, which inactivates NO and promotes endothelial dysfunction. Superoxide reacts with NO and forms peroxynitrite (ONOO^-^)—a powerful oxidant that further promotes apoptosis or necrosis. Peroxynitrite promotes loss of the bioactive NO and acts directly cytotoxically [8,9].

Histological materials have demonstrated high-density lipoprotein cholesterol (HDL-C) localization in calcific aortic valves already at initial stages of AS development [10]. The main HDL-C function is to promote the removal of cholesterol from the artery walls, to prevent the expression of cytokine-induced adhesion molecules in endotheliocytes, and to promote the production of nitric oxide [11]. Looking at the role of HDL-C in the pathogenesis of aortic stenosis, we found an HDL-C relationship with ROS [12]. HDL-C has antioxidant properties [13]. HDL-C is involved in reducing inflammation and apoptosis in cells [14]. Under the influence of myeloperoxidase (MPO), oxidized low-density lipoprotein (ox-LDL) is formed from low-density lipoprotein cholesterol (LDL-C) via ROS. Increasing ox-LDL levels cause HDL-C dysfunction and oxidized high-density lipoprotein (ox-HDL) formation, thus reducing HDL-C protection [12]. Therefore, it was interesting to discover and study the relationship of HDL-C with the biomarkers of oxidative stress by examining patients’ blood plasma and serum in the control group and separately in three aortic stenosis severity grades. From the oxidative stress biomarkers, we selected myeloperoxidase (MPO, a leukocyte-derived enzyme, which catalyzes the formation of reactive oxygen species and is an index of oxidative stress) and thioredoxin reductase 1 (TrxR1, a cytosolic enzyme that plays a central role in controlling cellular redox homeostasis) [15,16]. MPO, a pro-oxidant enzyme that is associated with both inflammation and oxidative stress, is located in neutrophilic leukocytes [14]. MPO released from activated neutrophils and macrophages worsens HDL-C protective properties [13]. The thioredoxin (Trx) system consists of nicotinamide adenine dinucleotide phosphate (NADPH), thioredoxin reductase (TrxR), and Trx, and it protects against oxidative stress. Trx is dependent on TrxR action to reduce peroxide, ribonucleotide and methionine sulfoxide, thus affecting metabolism and reducing inflammation, proliferation, and apoptosis [17].

The aim of the present study was to evaluate the role of HDL-C in calcific aortic stenosis development and the HDL-C relationship with biomarkers of oxidative stress in blood serum and plasma in all three aortic stenosis severity grades compared to the control group.

## 2. Materials and Methods

### 2.1. Patient Populations

The clinical study was approved by the Riga Stradins University Ethics Committee on Research on Humans. The study protocol conforms to the Ethical Guidelines of the 1975 Declaration of Helsinki (Approval number 12.09.2013/11, approved on 12 September 2013). A total of 102 patients volunteered according to the inclusion and exclusion criteria and were divided into two main groups: the control group and the AV stenosis group, according to the 2012 European Society of Cardiology and the European Association for Cardio-Thoracic Surgery Guidelines for the Management of Valvular Heart Disease [18]. As shown in Table 1, 28 (27%) men and 74 (73%) women were included in the study. Written informed consent to participate in the study was obtained from each individual enrolled in this study.

Individuals in the control group were included according to the echocardiographically confirmed healthy aortic valve. Exclusion criteria for both the control and the AS groups were the following: connective tissue diseases, infectious diseases, oncological diseases, diabetes mellitus, thyroid disfunction, severe, moderate, and uncontrolled arterial hypertension, history of acute coronary syndrome and manifested coronary heart disease, left ventricular systolic dysfunction with reduced ejection fraction (EF) below 50%, cerebral infarction and transient ischemic attack, echocardiographically confirmed cardiomyopathy, visual AV sclerosis, pathologies of other valves, and no lipid lowering therapy used. The exclusion criterion in the patient group with aortic valve stenosis was congenital (for example, bicuspid aortic valve) and rheumatic aortic valve damage.

Ankle-brachial index (ABI) was determined for all study subjects before inclusion in the study. The obtained results were evaluated according to the recommendations of the American Heart Association [19]: ≥1.4 indicates calcified, non-compressible arteries; 1.0–1.39 normal ABI; if there is claudication, then an exertional test is performed; 0.91–0.99 possibly, there is a peripheral arterial disease; <0.9 there is a peripheral arterial disease; ≤0.5 severe ischemia, and <0.4 critical ischemia. The intima-media thickness (IMT) in the common carotid artery was determined in all individuals involved in the study, which was considered normal if <0.9 mm.

### 2.2. Stenosis Assessment

Echocardiography with data saving was performed for all persons prior to inclusion in the study, using the GE VIVID 7 Dimension (GE Medical Systems, Milwaukee, Wis., USA) and Philips IE 33 (Philips Ultrasound, Inc., Bothell, WA, USA) echocardiography devices. Each echocardiography examination was evaluated by two echocardiography specialists. Patients with aortic valve stenosis were divided into three subgroups (mild, moderate, and severe), depending on the severity of the AS, according to the criteria of the 2012 European Society of Cardiology and the European Association for Cardio-Thoracic Surgery Guidelines for the management of valvular heart disease criteria [18]:aortic jet velocity—Vmax (m/s),mean pressure gradient—PG mean (mmHg),aortic valve area—AVA (cm^2^),indexed aortic valve area—indexed AVA (cm^2^/m^2^).

Severe AS: Vmax >4 m/s, PG mean >40 mmHg, AVA <1.0 cm^2^, indexed AVA <0.6 cm^2^/m^2^; moderate AS: Vmax 3.0–4.0 m/s, PG mean 20–40 mmHg, AVA 1.0–1.5 cm^2^, indexed AVA 0.60–0.85 cm^2^/m^2^; mild AS: Vmax 2.5–2.9 m/s, PG mean <20 mmHg, AVA >1.5 cm^2^, indexed AVA >0.85 cm^2^/m^2^.

### 2.3. Laboratory Assay

Study subjects’ venous blood samples were collected after overnight fasting, centrifuged, and stored at −80 °C. High-density lipoprotein cholesterol (HDL-C) was determined using commercially available tests (D-HDL, Siemens Healthcare Diagnostics Inc., Tarrytown, NY, USA) using the direct method. HDL-C was determined using the Siemens Advia 1800 analyzer (Siemens Healthcare Diagnostics Inc., Tarrytown, NY, USA) according to the manufacturer’s protocol.

For the determination of TrxR1 in the blood plasma, the human thioredoxin-1 ELISA Assay Kit (Prod. #RAB1756/Lot #0522F2032, Sigma-Aldrich, Inc., St. Louis, MO, USA) was used; for the determination of MPO in the blood plasma, the human myeloperoxidase ELISA Assay Kit, Item No. 501410, Cayman Chemical Company, Ann Arbor, MI, USA was used. The results were obtained using an Infinite 200 PRO multimode reader (Tecan Group, Mannedorf, Switzerland) and a Multiskan Ascent microplate reader (Thermo Labsystems, Helsinki, Finland). The procedures were performed according to the protocol of the ELISA kit manufacturer.

Concentrations of lipids, glucose, and other routine blood biomarkers were analyzed by standard methods.

### 2.4. Statistical Analysis

All graphical images, calculations, and statistical analyses included in the study were performed using IBM SPSS (Statistical Package for the Social Sciences) Statistics 23 (IBM Corp., Armonk, NY, USA) and GraphPad Prism 7.0 software (GraphPad Software, San Diego, CA, USA) as well as Microsoft Excel 2013 (Microsoft, Redmond, WA, USA).

Normal distribution of data was tested by Brown–Forsythe and Bartlett tests or by the Kolmogorov–Smyrnov one-sample test. If the sample dispersion corresponded to the normal distribution, they were showed as the mean value (M) and standard deviation (± SD). Otherwise, the data are displayed as a median and interquartile range (IQR). In all cases, the Benjamin, Krieger, and Yekutieli statistical analysis method was used as a post-hoc analysis. The geometric mean and geometric standard deviation were used to reflect the results and data which by their distribution were more in line with logarithmic data distribution. The *p*-value of less than 0.05 (*p* < 0.05) was considered statistically significant for all used statistical tests. The method of correlation analysis was used to study the relationship of quantitative variables. Depending on the distribution of the data, we used a parametric (Pearson) or non-parametric (Spearman) correlation analysis, using GraphPad Prism 7.0 software. 

## 3. Results

### 3.1. Baseline Characteristics of Study Patients and Control Group Individuals

The basic data of the subjects included in the study are presented in Table 2. The average age of the patients in all aortic stenosis groups and in the control group was similar, and the mean body mass index (BMI) did not differ between the groups. The mean values of triglycerides and low-density lipoprotein cholesterol (LDL-C) were not statistically different between the stenosis groups and the control group. The groups were similar for the mean values of the ejection fraction (EF) determined by the Simpson’s method and the stroke volume (SV) measured by the left ventricular outflow method as well as according to the inclusion and exclusion criteria.

Focusing on the strict clinical and echocardiography exclusion criteria allowed us to choose the most appropriate study groups. Although the number of patients in the subgroups was limited, the results from the statistical analysis of the data revealed significant *p*-values ranging from *p* < 0.05 to *p* < 0.0001.

### 3.2. High-Density Lipoprotein Cholesterol (HDL-C) Level Differences between the Patient Groups

HDL-C is a very important anti-atherogenic factor that plays an important role in maintaining a positive optimal balance between other lipoproteins with atherogenic effects. In this study, we revealed that the HDL-C level in AS patients compared to controls substantially decreased (*p* < 0.0001) (Figure 1) and that this decline was observed in similar manner in all three AS severity groups (Figure 2). The results showed a significant difference in the HDL-C level between the mild stenosis (*p* = 0.018), moderate stenosis (*p <* 0.0002), and severe stenosis groups (*p* = 0.004) compared to controls.

Additional analysis was done to evaluate possible differences in HDL-C levels between male and female individuals in the AS group and the control group (Figure 3). The obtained results confirm a significant difference between women and men in the aortic valve stenosis group (*p* = 0.012) and in the control group (*p* = 0.0006). In both the control group and the stenosis group, the HDL-C level was higher in women than in men.

The HDL-C level was lower in the AS group both for women and men compared to the control group, as shown in Figure 3. The HDL-C level in women significantly differed (*p* = 0.0001) between the AS group and the control group, and also for men (*p* = 0.049), but for men this occurred with less statistical significance than for women. This can be explained by the small number of men in the groups.

Regarding the severity of AS, z-score analysis revealed that the HDL-C concentration decline was substantial (Figure 4A,B) and similar in all three AS severity groups in women: mild *p* = 0.008, moderate *p* = 0.008, and severe *p* = 0.005. However, in men, the HDL-C decrease was detected in the moderate group (*p* = 0.012) in comparison to controls. This can be explained by the small number of men in the groups.

### 3.3. Thioredoxin Reductase-1(TrxR1) Level Differences between the Patient Groups

We analyzed thioredoxin reductase-1 (TrxR1) in the blood plasma. When determining the TrxR1 level in the plasma in the control group subjects and in the patients with aortic valve stenosis, we obtained statistically significantly higher levels in the AS group (*p* = 0.0037).

When analyzing differences in the TrxR1 level between the control group and the three severity grades of the aortic valve stenosis, we obtain statistically significantly (*p* = 0.0001) higher TrxR1 levels in patients with mild aortic valve stenosis and severe aortic valve stenosis (*p* = 0.047), see Figure 5.

A correlation analysis was performed using a linear regression line to search for possible correlations between TrxR1 and HDL-C, see Figure 6. No association was found between TrxR1 and HDL-C concentrations.

### 3.4. Myeloperoxidase (MPO) Level Differences between the Patient Groups

MPO levels were determined in the control group and in the patients with aortic valve stenosis in all three severity grades of the aortic valve stenosis. We obtained a result showing statistically significantly higher plasma myeloperoxidase levels in patients with aortic valve stenosis compared to the control group (*p* < 0.00003). After performing in-depth analysis and comparing myeloperoxidase plasma levels between severity grades of the aortic valve stenosis, we obtained statistically significant differences in all severity grades compared to controls (*p* = 0.024 mild stenosis; *p* = 0.002 moderate stenosis; *p* = 0.0015 severe stenosis), see Figure 7. The results showed that myeloperoxidase levels increased with the increase in the severity of the aortic valve stenosis and that they reached the highest level in the patient group with severe stenosis.

Using correlation analysis, we obtained a statistically significant negative (*p* = 0.047; *rp* = −0.28) correlation between MPO and HDL-C: the higher the MPO level, the lower the level of HDL-C, see Figure 8.

## 4. Discussion

In this study, we obtained data on HDL-C association with AS development, and we found that HDL-C levels differed statistically significantly between the control group and the AS group.

Moreover, persons included in the study did not receive lipid-lowering drugs, and individuals with subclinical atherosclerosis and obesity were excluded. HDL-C levels were lower in all AS patients compared to the control group individuals. Likewise, HDL-C levels were statistically significantly lower in each AS severity group as compared to the control group. Analyzing women and men separately, we obtained statistically lower levels of HDL-C in women in the AS group compared to women in the control group and also similar results for men. But the statistical significance for men was less marked than for women. This can be explained by the small number of men in the groups.

Taking into account the desired minimum HDL-C level for men (1.0 mmol/L) and women (1.2 mmol/L), we performed a z-score analysis separately for women and men in the control group and in all three AS severity grades. As a result of this analysis, we observed that the HDL-C levels for all women in the control group were greater than 1.2 mmol/L and that it was statistically significantly lower in all grades of AS severity. For all men, the HDL-C level in the control group was higher than 1.0 mmol/L and it was statistically significantly lower in the moderate grade of AS severity, where the number of patients was higher.

In order to understand why HDL-C levels are lower in AS patients, we performed an HDL-C evaluation in relation to MPO and TrxR1, the biomarkers of oxidative stress, as HDL-C is pathogenetically associated with oxidative stress and ROS. In a previous study, we did not get a statistically significant relationship between HDL-C and chemerin (inflammatory biomarker) and fibroblast growth factor-21 (FGF-21), which is able to improve endothelial function and inhibit apoptosis [20,21]. There was no relationship between HDL-C and the biomarkers of the inflammatory process.

The pro-oxidant enzyme MPO level differed statistically significantly between the control group and all three AS severity grades, increasing along with AS severity. An analysis of the correlation between MPO and HDL-C revealed a negative relationship: higher MPO levels were associated with lower HDL-C levels. This confirms the adverse effects of MPO where it increases the level of ox-LDL-C, causing HDL-C dysfunction and the production of ox-HDL-C, thereby reducing the protective action of HDL-C. Since the highest MPO level was found in the severe AS grade, this grade has consequently the lowest HDL-C level and protective ability.

Analyzing the antioxidant TrxR1, we obtained a statistically significant difference between the control group and the AS group, where TrxR1 levels were higher. Analyzing the severity grades of AS, we obtained statistically higher TrxR1 levels in mild and severe AS grades. A higher level of TrxR1 in the severe grade compared to moderate grade can be explained by myocardial hypertrophy. It has been demonstrated that myocardial hypertrophy contributes to TrxR1 expression [22] and/or heart failure; it has been proven that the more pronounced is the heart failure, the higher is the thioredoxin level [23].

The level of MPO and TrxR1 between the AS severity grades and the control group shows the role of oxidative stress in the development of AS. The relationship between MPO and TrxR1 with HDL-C reflects HDL-C participation in AS development and association with oxidative stress. This was proven definitely and convincingly in women. The inadequate number of men in our groups did not allow us to express the same statement. More research in larger groups of patients is recommended.

The data from the randomized prospective clinical trials SALTIRE and SEAS found no effect in the application of intensive lipid-reducing therapy in the case of AS. Atorvastatin 80 mg was given during the first study, whereas simvastatin/ezetimibe 40/10 mg per day was given for treatment of AS during the second study [24,25]. Similar results were obtained in the ASTRONOMER study, which concluded that treatment with rosuvastatin did not reduce mild and moderate AS progression [26], and detracted physicians and researchers’ thoughts from lipids and AS. However, when evaluating oxidative stress and HDL-C, we obtained results on the HDL-C relationship with the AS process. Perhaps, by creating a balance between ROS and antioxidants, we can improve the HDL-C protective action.

It is difficult to determine whether lower HDL-C levels in AS patients are due only to oxidative stress, which is present in all grades of AS. Additionally, there is a possibility that persons with lower HDL-C levels are predisposed to AS. Larger patient groups should be created to demonstrate HDL-C association with AS not only for women but also for men at all AS severity grades.

The role of HDL-C in AS patients has been previously assessed, and a small study (42 patients) demonstrated the importance of the HDL-C/TC ratio in the rate of AS progression [27]. In histological materials from AS patients, a smaller amount of HDL-C was found than in the valves of the control group [10]. The genetic variants and functional properties of HDL-C have been studied and the researchers concluded that the variants and properties did not affect the risk of AS formation [28].

In any case, the higher the level of HDL-C in AS patients, the better its protective properties. Since HDL-C (one of the lipid fractions) is considered to be a risk factor for cardiovascular disease, non-pharmacological recommendations for physical activity and proper nutrition are very topical.

At this time, only the study results for animals have been published, where the administration of ApoA1 reduced the manifestation of AS [29,30].

## 5. Conclusions

In this study, we justified HDL-C level association with AS development process. The results unequivocally substantiated the association between HDL-C and AS in all severity grades in women, but only in moderate AS for men, which we explained by the small number of men in the groups.

The obtained statistically significant correlation between HDL-C and MPO levels, as well as the concurrent decrease in the HDL-C level and increase in the TrxR1 level, indicate in general an HDL-C association with oxidative stress in AS patients.

## Figures and Tables

**Figure 1 medicina-55-00416-f001:**
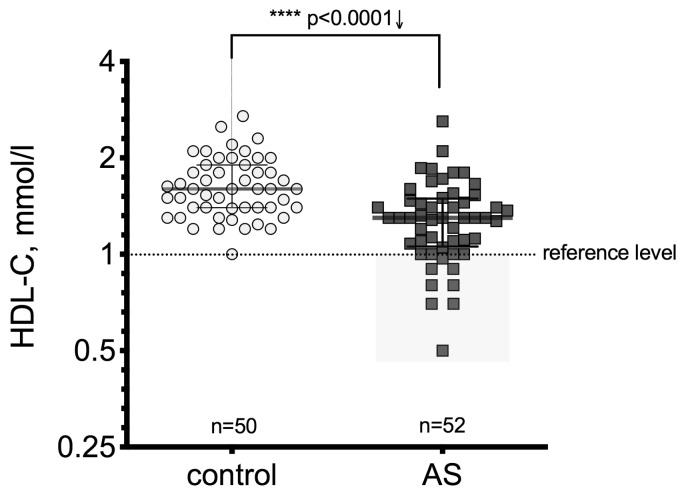
Serum high-density lipoprotein cholesterol (HDL-C) level in the control and aortic valve stenosis (AS) groups; the gray rectangle identifies cases below the reference level (considered as a risk zone). Asterisks show the level of statistical significance.

**Figure 2 medicina-55-00416-f002:**
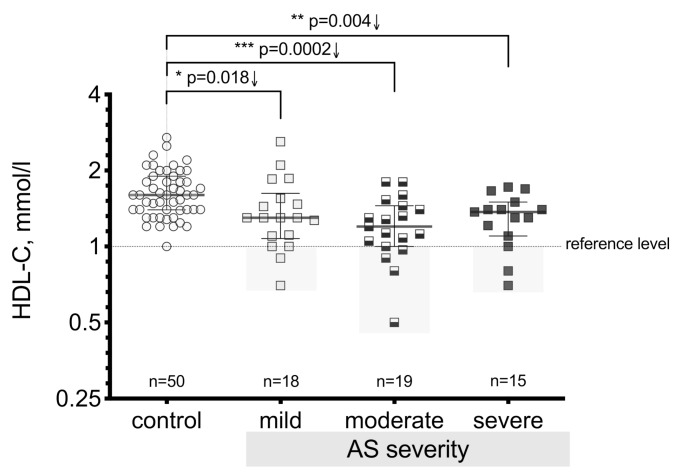
Serum HDL-C level in the control group and in all AS severity grades; the gray rectangles identify cases below the reference level (considered as a risk zone). Asterisks show the level of statistical significance.

**Figure 3 medicina-55-00416-f003:**
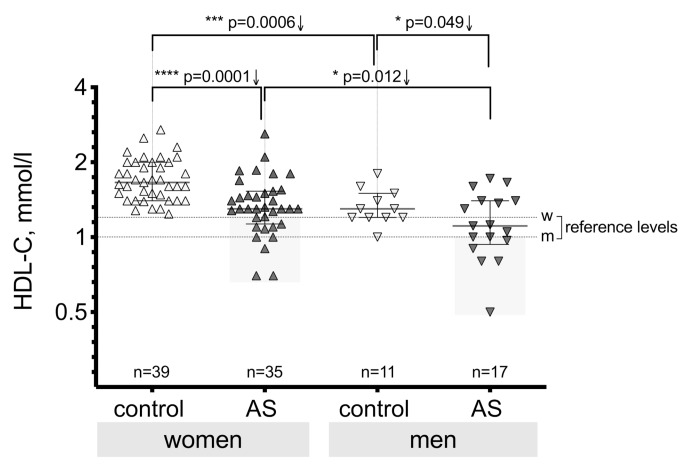
Serum HDL-C level in the control group and in all AS severity grades; HDL-C level gender differences (w—women, m—men); the gray rectangles identify cases below the reference level (considered as a risk zone). Asterisks show the level of statistical significance.

**Figure 4 medicina-55-00416-f004:**
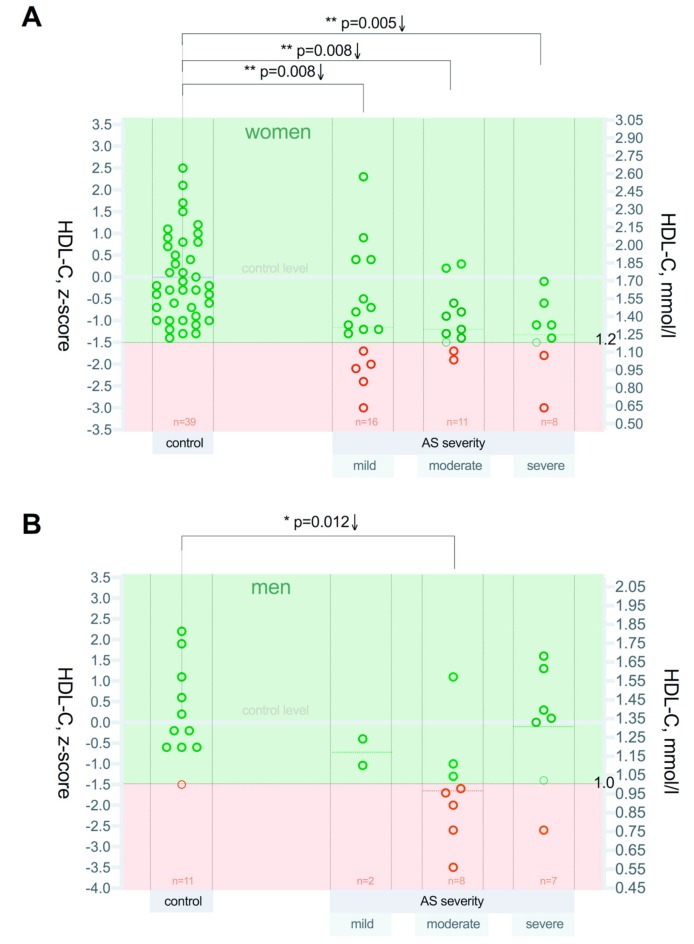
(**A**,**B**) Serum HDL-C concentration regarding AS severity, standard z-score representation. (**A**) serum HDL-C concentration in women and (**B**) serum HDL-C concentration in men. Asterisks show the level of statistical significance.

**Figure 5 medicina-55-00416-f005:**
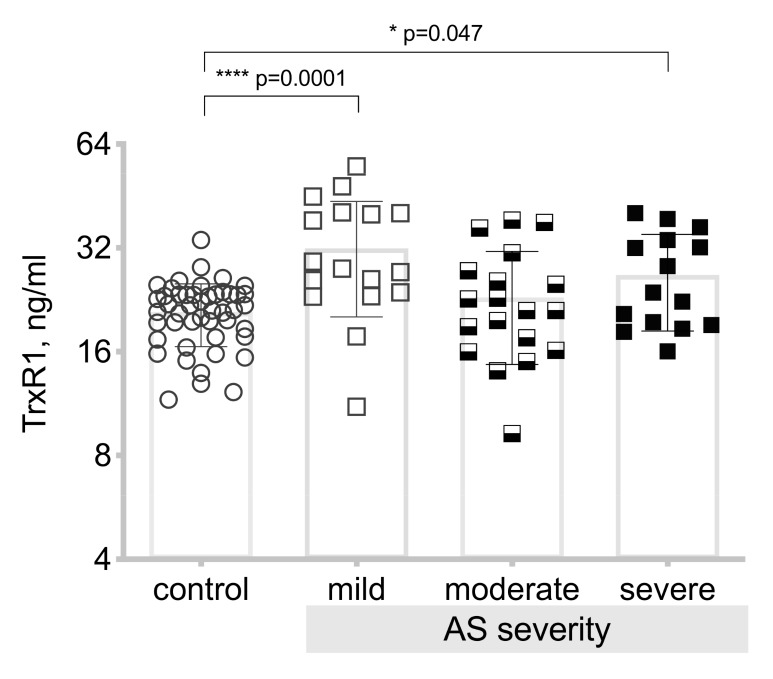
Plasma thioredoxin reductase 1 (TrxR1) level in the control and aortic valve stenosis groups. Asterisks show the level of statistical significance.

**Figure 6 medicina-55-00416-f006:**
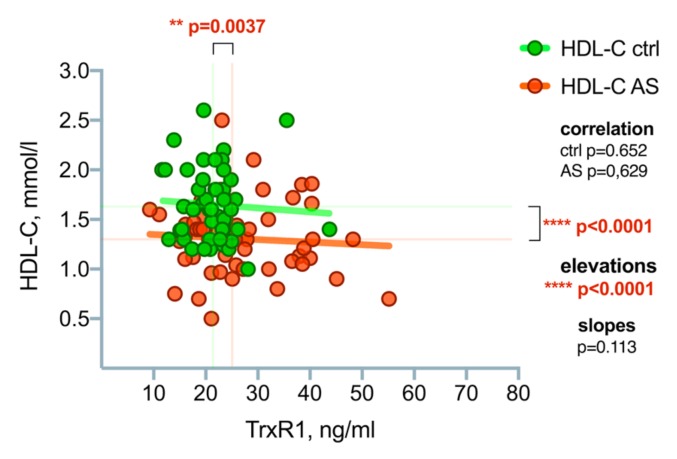
Correlation analysis between TrxR1 and HDL-C concentrations in the control group and aortic stenosis patients. Asterisks show the level of statistical significance.

**Figure 7 medicina-55-00416-f007:**
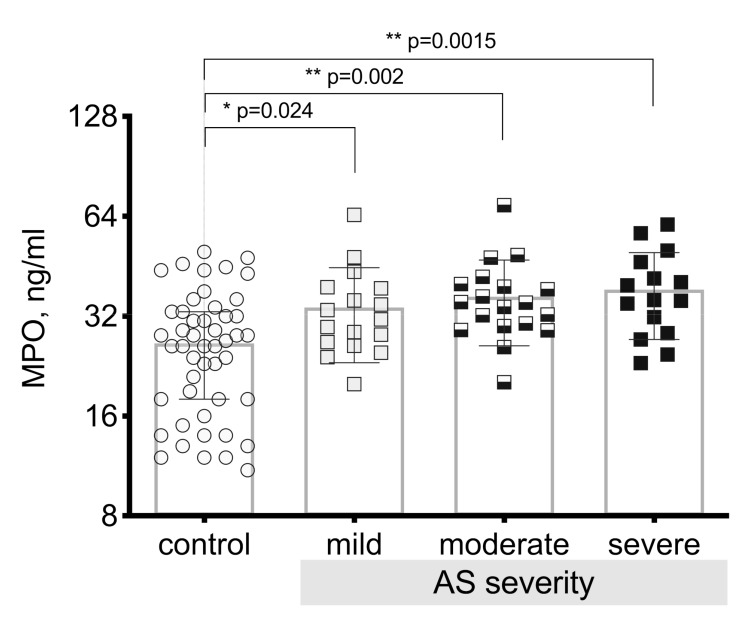
Plasma myeloperoxidase (MPO) level in the control group and in all AS severity grades. Asterisks show the level of statistical significance.

**Figure 8 medicina-55-00416-f008:**
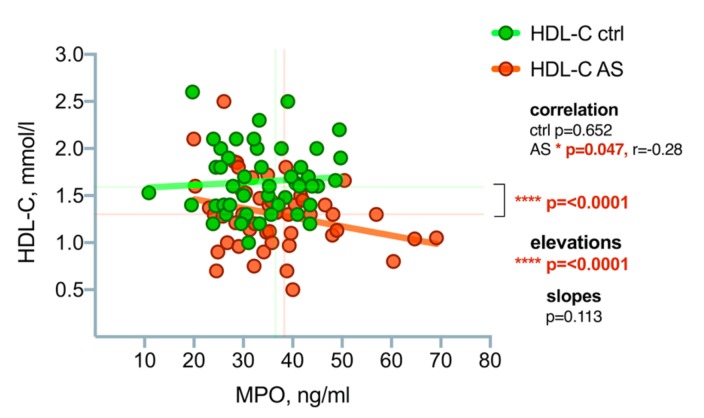
Correlation analysis between MPO and HDL-C concentrations in the control group and the aortic valve stenosis patients. Asterisks show the level of statistical significance.

**Table 1 medicina-55-00416-t001:** Baseline characteristics of study subjects.

		Control, *n* = 50	AV Mild Stenosis, *n* = 18	AV Moderate Stenosis, *n* = 19	AV Severe Stenosis, *n* = 15
**Gender, (%)**	Male Female	11 (22.0) 39 (78.0)	2 (11.1) 16 (88.9)	8 (42.1) 11 (57.9)	7 (46.7) 8 (53.3)
**Age, years**	Mdn (IQR)	64 (57–75)	71 (65–75)	74 (65–79)	65 (60–74)

**Table 2 medicina-55-00416-t002:** Basic data of individuals in the control group and patients in the aortic valve stenosis (AS) group.

			Control, *n* = 50	AV Mild Stenosis, *n *= 18	AV Moderate Stenosis, *n* = 19	AV Severe Stenosis, *n* = 15
**^1^ BMI**	M (± SD)		26.04 (4.31)	27.39 (3.10) *p* = 0.399	25.81 (4.58) *p* = 0.682	27.40 (3.18) *p* = 0.869
*p*-value vs. control						
**^2^ LDL-C, mmol/L**	M (± SD)		3.28 (1.18)	3.05 (0.97) *p* > 0.999	2.59 (0.92) *p* = 0.057	3.10 (1.12) *p* > 0.999
*p*-value vs. control						
**^3^ TG, mmol/L**	M (± SD)		1.47 (0.71)	1.64 (0.84) *p* = 0.406	1.11 (0.56) *p* = 0.178	1.27 (0.57) *p* = 0.406
*p*-value vs. control						
**^4^ TC, mmol/L**	M (± SD)		5.49 (1.28)	5.01 (1.34) *p* = 0.056	4.21 (1.18) ** *p* = 0.001	4.68 (1.08) * *p* = 0.016
*p*-value vs. control						
**^5^ SV, mL**		Mdn (IQR)	96.5 (90.0–106.3)	100.0 (90.0–110.0) *p* = 0.716	96.0 (88.0–100.0) *p* = 0.375	90.0 (88.0–95.0) *p* = 0.103
*p*-value vs. control						
**^6^ EF %**		Mdn (IQR)	63.5 (57.7–68.0)	60.0 (57.5–63.5) *p* = 0.347	61.0 (58.0–66.0) *p* = 0.981	60.0 (57.0–64.0) *p* = 0.347
*p*-value vs. control						
**^7^ SVI**		Mdn (IQR)	52.2 (46.3–59.1)	53.6 (49.6–60.2) *p* = 0.767	49.4 (47.4–52.1) *p* = 0.288	49.7 (42.9–52.7) *p* = 0.157
*p*-value vs. control						

^1^ BMI (body mass index)—weight in kilograms divided by the square of the height in meters, (kg/m^2^); ^2^ LDL-C—low-density lipoprotein cholesterol; ^3^ TG—triglycerides; ^4^ TC—total cholesterol; ^5^ SV—stroke volume, measured by left ventricular outflow method; ^6^ EF—ejection fraction, measured by Simpson’s method; ^7^ SVI (stroke volume index)—the relation between the stroke volume (SV) and the size of the person body surface area (BSA), mL/m^2^; asterisks show the level of statistical significance.

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
