# Peer review of "HDL-C Role in Acquired Aortic Valve Stenosis Patients and Its Relationship with Oxidative Stress"

_medicina, 2019, doi:10.3390/medicina55080416_

Round 1

Reviewer 1 Report

In discussion section , please discuss inflence of atorvastatin on lipids level and calcification biomarker in early stage of AS i.e. Aortic sclerosis/mild AS. 

Reviewer 2 Report

In the manuscript ID medicina-547840 entitled, "HDL-C role in acquired aortic valve stenosis patients and its relation with oxidative stress”, the authors aimed to evaluate the role of HDL-C in the calcific aortic stenosis development and the link of stenosis severity degrees to oxidative stress. The manuscript is well written and the presented data will add a new insight in acquired aortic stenosis.

There are just a few concerns:
Authors explained that the small sample size for men limits the statistical significance marked less than for women. However, as the data in Figure 3 indicates, women tend to have higher levels of HDL cholesterol than men do. The potential cause and consequences of gender differences should be at least discussed. In this regard, I am wondering if there are any gender difference of the expression levels of TrxR1 and MPO or the correlations with HDL-C level.

In abstract, one additional statement describing the rationale to focus on HDL-C in the background section would be appreciated to follow the concept of study.

In Table 2, add asterisk mark indicating statistically significant might be appreciated by readers.

In multiple figure legends, I wasn’t sure what “gray rectangles mark cases in risk zone” really meant. It would be great if authors can explain it a little more.

It was stated that “No association was found between TrxR1 and HDL-C concentrations.” in the results. On the contrary, it was stated that “Obtained statistically significant correlations between HDL-C levels and TrxR1…” in the conclusions. Which statement is correct? In this regard, a few more details might be needed to clarify in the legends of Figures 6 and 8.

There is a typo in page 4 on line 137 (Thyredoxin-1).
